# Evaluation of the Climate Impact and Nutritional Quality of Menus in an Italian Long-Term Care Facility

**DOI:** 10.3390/nu16172815

**Published:** 2024-08-23

**Authors:** Andrea Conti, Annalisa Opizzi, Jefferson Galapon Binala, Loredana Cortese, Francesco Barone-Adesi, Massimiliano Panella

**Affiliations:** 1Department of Translational Medicine, Università del Piemonte Orientale, 28100 Novara, Italy; 20022851@studenti.uniupo.it (A.O.); 20042715@studenti.uniupo.it (J.G.B.); 20042441@studenti.uniupo.it (L.C.); francesco.baroneadesi@uniupo.it (F.B.-A.); massimiliano.panella@med.uniupo.it (M.P.); 2Doctoral Program in Food, Health, and Longevity, Università del Piemonte Orientale, 28100 Novara, Italy; 3Anteo Impresa Sociale, Via Felice Piacenza 11, 13900 Biella, Italy

**Keywords:** nursing home, carbon footprint, healthcare foodservice, planetary health diet

## Abstract

Global warming poses a significant threat to our planet, with the food sector contributing up to 37% of total greenhouse gas emissions. This study aimed to assess the climate change impact and healthiness of menus in a long-term care facility in Italy. We analyzed two 28-day cyclical menus using the carbon footprint (CF) and the Modified EAT-Lancet Diet Score (MELDS) to evaluate adherence to the Planetary Health Diet (PHD). Monte Carlo simulations were employed to explore 20,000 daily menu permutations. Results showed that the mean GHGEs of spring/summer and autumn/winter daily menus were 2.64 and 2.82 kg of CO_2_eq, respectively, with 99% of menus exceeding the 2.03 kg of CO_2_eq benchmark. Only 22% of menus were adherent to the PHD, with MELDSs ranging from 12 to 29. A strong inverse association between the CF and adherence to the PHD was observed. These findings suggest significant potential for reducing the CFs of meals served in nursing homes while promoting adherence to a planetary diet, presenting an opportunity to set new standards in caregiving and environmental sustainability.

## 1. Introduction

In the last several years, global warming has become a prominent threat to our planet, being related to several harmful effects like increasing temperatures, rising sea levels, and melting ice caps [1]. Human activities, through continuously increasing greenhouse gas emissions (GHGEs), represent major contributions to this phenomenon [2]. In the actual debate on climate change the food sector plays a pivotal role, due to its presence and relevance in all populations. Indeed, food production is responsible for up to the 37% of the total GHGEs [3], equal to 18 Gigatonnes of carbon dioxide equivalents (CO_2_eq) [4]. Furthermore, the food sector is not only contributing to global warming but, at the same time, it is threatened by the consequences of the latter, with serious implications for nutrition security, livelihoods, and general well-being [5].

Among the various strategies proposed to address this global challenge, the adoption of healthy and sustainable food models has been identified as one of the most important interventions [5]. Sustainable foods prioritise environmental health, social equity, and economic viability throughout their lifecycle [6]. This concept encompasses practices mitigating the impact on the environment, such as reducing greenhouse gas emissions, conserving water, and promoting biodiversity. Ultimately, the goal of sustainable food is to foster a harmonious relationship between food production, consumption, and the planet’s ecosystems [7].

In this regard, the EAT-Lancet Commission has proposed the Planetary Health Diet (PHD) to promote both human and environmental health [8]. This dietary model, mainly based on vegetables, has been designed to be translated to different populations, while respecting their eating habits and cultural traditions. Such flexibility allows for the intake of all necessary macronutrients to ensure a proper diet while limiting GHGEs.

The recent awareness of food sustainability has been reflected in an important increase, in the last several years, in the amount of research on this topic [9]. While a similar trend can be observed among the literature on healthcare services sustainability, the number of studies explicitly investigating the intersection of food and healthcare sustainability remains limited and anecdotal [10]. Indeed, while some studies focused on production and preparation processes [11] or on the hospital setting in general [12], as far as the authors of this paper know, currently, there are no studies assessing the climate impact of food services in long-term care (LTC) facilities such as nursing homes. This is particularly concerning, especially considering that this sector not only accounts for more than 1.3 and 3.4 million residents in the United States and in the European Union, respectively, but also it is continuously growing [13,14]. Moreover, LTC stays are substantially longer than hospital ones, with subjects exposed for longer amounts of time to the served food and potentially having more benefits from its improvement. We hypothesise that, in line with evidence from general populations [15], food in the LTC sector can be improved to benefit both the residents and the environment.

Our study aims to address this knowledge gap by evaluating dietary practices in LTC facilities, specifically assessing the climate change impact and healthiness of menus proposed to the residents of an Italian nursing home.

## 2. Materials and Methods

This study was conducted in the “Belletti Bona” nursing home (Biella, Piedmont, Italy). This facility, managed by the not-for-profit company Anteo Impresa Sociale, has been selected as its characteristics make it representative of Italian nursing homes. Belletti Bona can host up to 144 residents, and it is one of the largest nursing home of Biella. It hosts a large variety of residents, characterised by levels of independence and needed care [16]. Moreover, the majority of residents are aged 65 and older, with most stays co-funded by the regional government, aligning with the national context [17].

Dishes for the residents are prepared in the internal kitchen. The foodservice operates on two 28-day cyclical menus (one for the spring/summer period and another for the autumn/winter one). Five different meals (breakfast, a mid-morning snack, lunch, a mid-afternoon snack, and dinner) are served daily. Table 1 shows the structure of each meal. In addition, the foodservice provides customized options during the major holidays (e.g., Christmas, Easter, etc.), as well as special dishes for residents with specific dietary needs (e.g., because of diabetes, renal failure, etc.).

The details regarding the proposed menus were acquired through the food management software (NovaPortal 2022.04.28, Nova Srl, Bassano del Grappa, Italy, 2023), used by Anteo Impresa Sociale to manage the food chain across most of its facilities. Specifically, we obtained the scheme of the proposed servings for each menu and we extracted the recipes for each dish, including the lists of ingredients along with their respective weights per portion. A typical 28-day cyclical menu includes a total of 150 different dishes, which can be differently combined among each other as the residents can choose among different options (Table 1).

To estimate the climate change impact, we calculated the average carbon footprint (CF) associated with each dish. The CF, namely “the total amount of carbon dioxide emission that is directly and indirectly caused by an activity or is accumulated over the life stages of a product” [18], is an internationally adopted proxy for estimating the contribution of human processes to global warming. The CF is commonly used in studies on the food service sector [19] and is now being used also across healthcare services [20]. According to the available literature, we expressed the CF as kilograms of CO_2_eq [21]. The CF estimates of the different ingredients of dishes were gathered from the scientific literature. In detail, most of the estimates were obtained from an Italian database [22], in which life cycle assessment (LCA) studies consistent for food processing, distribution, and retailing were summarized. Furthermore, additional CF estimates of fresh ingredients, frozen products, and wine were obtained from other sources [23,24,25,26]. Our study considered the CF generated during the production, distribution, and retailing phases of the ingredients. In addition, it is worth mentioning that we did not consider salt and pepper, because of their minimal contribution to the overall weight of the proposed servings.

The adherence of the daily menus to the PHD was assessed using the Modified EAT-Lancet Diet Score (MELDS) developed by Stubbendorff et al. [27], which assigns a score from 0 to 3 (for low to high adherence to the EAT-Lancet diet food component target, respectively) to 14 different food categories. Similar to what we did for CF estimation, we considered the basic ingredients and their proportion for each dish to calculate the adherence score.

We considered 2.03 kg of CO_2_eq/person/day as a benchmark standard for the CF. This value represents the daily average CF of a complete adherence to the PHD, specifically estimated in the Italian population [28]. Benchmark for the MELDS was set to 23 in accordance with the original tool [27]. Detailed information about estimates for each dish is available in the Appendix A.

Finally, we used Monte Carlo simulations to explore the whole amount of food offerings in the spring/summer and autumn/winter 28-day cyclical menus, drawing on 20,000 daily menus from all the possible permutations of dishes available during the two periods. We then plotted values of the CF and the MELDS for each daily menu to evaluate how close they were to the two benchmarks. Statistical analysis was performed using Stata software (StataCorp. 2021. Stata Statistical Software: Release 17. College Station, TX, USA: StataCorp LLC).

## 3. Results

Mean GHGEs of spring/summer and autumn/winter daily menus were 2.64 and 2.82 kg of CO_2_eq, with 95% of permutations ranging between 2.18 and 3.17 kg of CO_2_eq, and 2.36 and 3.29 kg of CO_2_eq, respectively (Figure 1). Figure 2 represents the adherence of menus to the EAT-Lancet Planetary Health Diet during the two considered periods. Both the spring/summer and the autumn/winter menus ranged from a MELDS of 12 to 29, with a median of 20 and 21, respectively.

The association between the MELDS and the carbon emissions is reported in Figure 3. The large majority of the menus (99%) were above the 2.03 kg of CO_2_eq cutoff, and only a few (22%) were adherent to the PHD (i.e., they had a MELDS of at least 23).

Figure 4 shows a strong, inverse association between the CF and the adherence to the PHD, through the whole range of MELDSs. This phenomenon was present during both the spring/summer and autumn/winter periods.

## 4. Discussion

Our study aimed to explore the impact on climate change and the healthiness of the food served in an LTC facility, ultimately prompting research in this setting.

As a major result, we found a substantial variability in the CFs and MELDSs among possible menus, with most of them not reaching the CF gold standard. This result aligns with those from previous studies conducted in general hospital settings [12]. Moreover, adherence to the PHD was satisfactory in just half of the considered menus. From this perspective, our findings suggest not only the fact that the proposed menus are largely not sustainable, but also that there is room for their improvement. Furthermore, we found that a high MELDS was strongly associated with a reduction in the CF, corroborating the hypothesis that a shift to a PHD-compliant diet could lead to both environmental and health benefits, confirming our hypothesis.

A recent study [28] evaluated the current diet of the Italian general population, estimating that the adoption of a PHD-based diet would cut the amount of GHGEs in half. The same study highlights, also in accordance with the FAO [29], the dominant role of meat and dairy products in determining CFs. In addition to reducing GHGEs, introducing a PHD-compliant diet could also benefit LTC residents, translating into a reduction in the amounts of red meat and dairy products. Indeed, it has been observed that a frequent and excessive consumption of such foods in older people is associated with frailty [30]. Furthermore, this dietary pattern is linked to an increased susceptibility to hip fractures and to a deterioration in cognitive function [31,32]. Therefore, a diet based on the scheme proposed by the EAT-Lancet Commission can be of pivotal importance for frail subjects such as older people.

In addition to the above-mentioned considerations, economic aspects should also be taken into account. Indeed, it has been estimated that the adoption of a planetary diet would not be economically sustainable for several countries, especially for the low- and middle-income ones. For example, in Europe a PHD-complaint diet would cost about 2.8$ per day, 60% more than the minimum cost of a nutrient adequate one [33]. With regards to the LTC sector, foodservice expenses represent only 15% of the total running costs. More specifically, food supplies account for 39% of these dietary expenditures [34]. Hence, the implementation of a PHD-compliant diet might lead to a limited, although significant, increase in direct costs. While the available literature assesses the economic aspects of planetary diet adoption on a individual level [33], as far as the authors of this paper know, there are no similar studies conducted in collective foodservices. Therefore, it is not possible to currently estimate whether a fully PHD-compliant diet could be financially sustainable for the LTC sector.

It should be also taken in consideration that the PHD is a general dietary framework designed to be adapted and translated to local habits and traditions. Notably, a Mediterranean diet aligned with PHD principles for the Italian context was recently proposed [35,36]. This version of the PHD assumes a decreased consumption of resource-intensive foods like chicken, fish, and eggs compared to the original PHD framework, and subsequently, a further lower CF. Despite the authors being cautious to propose this model to the general population, this was a first example of the translation of the PHD to a specific country, while preserving the population background and culture. Furthermore, it has been estimated that moving from the current Italian food consumption pattern to a PHD-compliant one could lead to more than a 50% reduction in food-related GHGEs, namely 1000 kg of CO_2_eq per capita yearly [37]. Since, in Italy, more than 360,000 older people live in LTC facilities [38], this transition might potentially lead to a relevant overall reduction of GHGEs. While previous studies highlighted several barriers for the implementation of healthy diets, it should be considered that, usually, a relevant role is played by social and environmental factors such as social stigmatization, media influence, and the convenience and ease of unhealthy foods [39]. Since in the LTC sector these factors can be relatively easy driven at least with the support of trained healthcare workers [40], we suggest that such a setting could be fertile soil on which it might be possible to effectively start the implementation of healthy and environmentally friendly diets. This context-specific strategy could be of particular importance for cultures in which food plays a relevant role, such as, for example, in Italy and other Mediterranean countries [41]. However, it is worth mentioning that the PHD was originally designed for the healthy average adult, and it could not fully provide the nutritional requirements of specific categories of people such as adolescents, pregnant women, and older persons. In this regard, the adaptation of the PHD to population-specific nutritional needs remains one of the main challenges that still need to be addressed to achieve a comprehensive sustainable dietary approach [5].

Our study presents a number of limitations. First, the study was conducted in a single facility. In Italy, each seasonal menu is designed by the facility’s managing institution considering organizational, economic, and regulatory aspects, but has to be approved by the province-based local health authority (ASL, Azienda Sanitaria Locale). Since in Italy there is no a national guideline on nutrition in LTC settings, each one of the 110 different ASLs might follow different criteria to approve the foods offered. Despite analyzing a relevant number of different permutations, our results might not be generalized to all Italian facilities, and therefore, this study should be viewed as a pilot study. Indeed, we aim to scale-up this approach to the majority of the facilities managed by Anteo, which are distributed across several ASLs, to achieve a more comprehensive assessment of the foods offered in LTC [42].

Second, our study was carried out considering all the possible menu permutations based on dishes available in a specific period. As mentioned above, residents can choose between four different recipes for the first and second course, and between three different proposals for side dishes and dessert. Personal preferences and cultural backgrounds are well known determinants of food preferences [43], as well as the changes in nutritional habits in older ages [44]. However, as our study did not weight the different permutations considering residents’ choices, our results should be interpreted as a comprehensive analysis of the facility’s food offerings rather than an assessment of individual choices.

Third, we did not take in consideration the roles of on-site cooking and of food waste. In our specific case, the facility used had several appliances with different energetic performances, not allowing us to accurately calculate the electricity and gas usage. However, our study accurately assessed the food production, distribution, and retailing phases, which are responsible for 84% of the kg of CO_2_eq produced by catering services [45]. To overpass the abovementioned limitations, future research might consider multiple facilities, as well as other phases of the foodservice chain and residents’ preferences. Food waste accounts for an estimated 30% of global food production along the supply chain [29]. However, in hospital settings, up to 65% of food is wasted after being served to patients [46]. Therefore, we believe that this specific aspect should be assessed separately from the climate change impact of menus. Food waste in the healthcare sector is a complex phenomenon, driven by different determinants. For example, inappropriate dining times, large portions, and poor food presentations can contribute to the food waste in this setting [46,47]. Recent reviews addressed this topic in hospitals [48] and general hospitality services [49]; however, despite the growing interest, only a few studies have been specifically conducted in the LTC sector [50,51].

It is important to note that our study focused solely on the climate change impact through the estimation of GHGEs. While this approach has been widely adopted in recent years and facilitates comparison with other research [12,19,20], it represents a mono-dimensional perspective that may not capture the full spectrum of environmental impacts associated with the assessed human activities. A more comprehensive understanding would require consideration of additional factors such as particulate matter, nitrogen, and water footprints [52]. These additional metrics would provide a multi-dimensional view of the environmental impact of the activities under investigation.

To the best of our knowledge, our study is the first one specifically assessing the climate change impact and healthiness of the food in a nursing home. Considering the growing importance of the LTC sector worldwide and the subsequent related food production, future research on this topic seems warranted. Therefore, future studies might explore the individual food choices made by residents to provide a more accurate and realistic assessment, also considering individual characteristics such as gender, race, age, and socioeconomic status. This global approach could also include analysing dietary practices in LTC facilities across diverse countries, healthcare systems, and cultural contexts, allowing for a comprehensive understanding of how sustainability and health intersect with food choices in elderly care worldwide.

## 5. Conclusions

Our findings suggests that there is significant potential to reduce the CFs of the meals served to nursing home residents and to promote the adherence to a planetary diet at the same time. Current practices may not fully align with the guidelines set by the planetary diet proposed by the EAT-Lancet Commission. Adopting these guidelines can not only lead to a more sustainable food production and consumption pattern, but can also offer health benefits. Embracing this change presents an opportunity to set a new standard in caregiving while making a positive impact on the environment.

## Figures and Tables

**Figure 1 nutrients-16-02815-f001:**
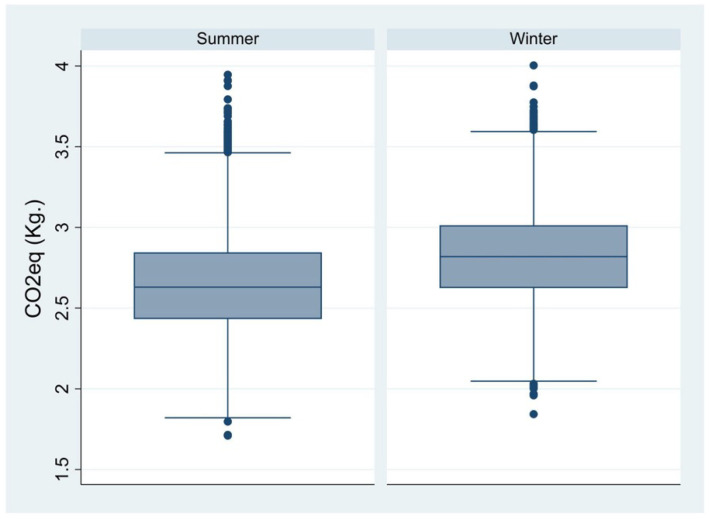
Daily GHGEs of the two seasonal menus.

**Figure 2 nutrients-16-02815-f002:**
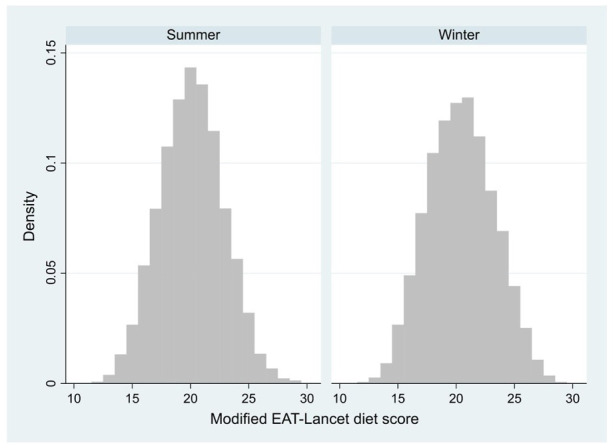
Adherence to the EAT-Lancet Planetary Health Diet.

**Figure 3 nutrients-16-02815-f003:**
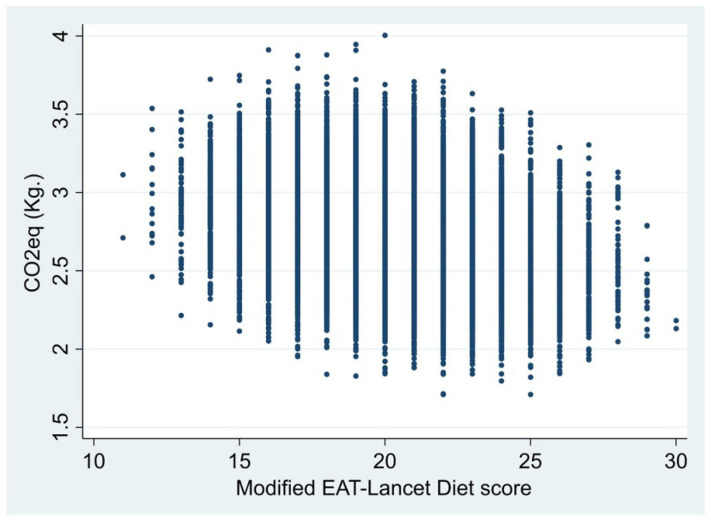
Distribution of the sample of 20,000 permutations.

**Figure 4 nutrients-16-02815-f004:**
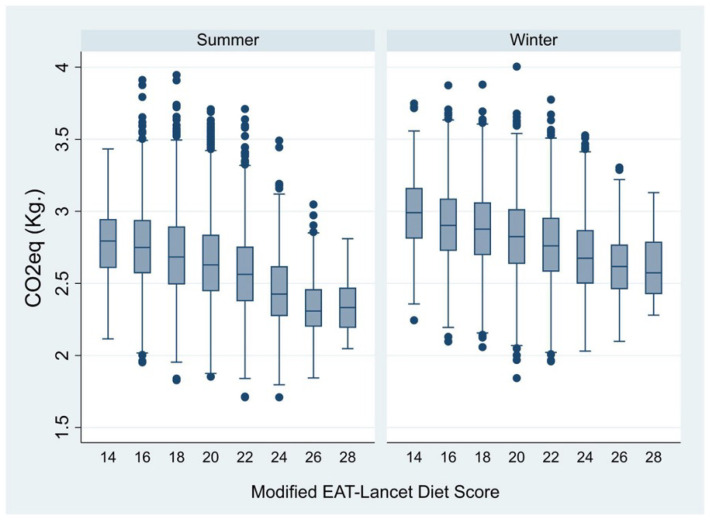
Association between GHGEs and adherence to the EAT-Lancet Planetary Health Diet.

**Table 1 nutrients-16-02815-t001:** Meal structure.

Meal		Number of Available Choices
Breakfast		3
Mid-morning hydration ^1^		1
Lunch	First course ^2^	4
	Second course	4
	Side dish	3
	Dessert/Fruits	3
Afternoon snack ^3^		3
Dinner	First course ^2^	4
	Second course	4
	Side dish	3
	Dessert/Fruits	3

^1^ 150 mL of fruit juice. ^2^ This serving is also available in a modified consistency (smoothie meal) for dysphagic patients. ^3^ Confectionery products.

## Data Availability

The dataset is available on reasonable request from the authors. The data are not publicly available due to internal policies.

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
