# Peer review of "Evaluation of the Climate Impact and Nutritional Quality of Menus in an Italian Long-Term Care Facility"

_nutrients, 2024, doi:10.3390/nu16172815_

Round 1

Reviewer 1 Report

Comments and Suggestions for Authors

This is study is competently conducted. I have just a few considerations to raise. 

1. Please describe the site selection process and take care to offer a justification for the agency studied. Add this in a new paragraph in Materials and Methods.  

2. Can we know more about the study participants, such as gender, race, etc.? Are the full gamut of characteristics featured here?

3. There are some typos in the manuscript that need correcting. 

I wish you well in your work with this paper. 

Comments on the Quality of English Language

A few typos need correction. 

Author Response

We would like to thank the reviewer for the effort in reviewing our manuscript. Below, our answers. Changes in the manuscript are highlighted in red. 

1. Please describe the site selection process and take care to offer a justification for the agency studied. Add this in a new paragraph in Materials and Methods.

We improved the materials and methods section according to this suggestion.

2. Can we know more about the study participants, such as gender, race, etc.? Are the full gamut of characteristics featured here?

Since the study was conducted on the proposed menus, there are no human participants involved in our study. However, we added brief information about residents living in the selected facility (please see in Materials and methods, also according to your previous comment).

3. There are some typos in the manuscript that need correcting. 

We revised the entire manuscript.

Reviewer 2 Report

Comments and Suggestions for Authors

Dear authors, thank you for submitting the manuscript "Evaluation of the climate impact and nutritional quality of menus in an Italian long-term care facility". I enjoyed reading your article and here is the feedback:

-Please expand the introduction, it is very short, only 3 paragraphs. You can expand the area of sustainable food such as mentioning different types of food vegetables easily growth in different parts of the world due to the different climates.

-Mention your hypotheses at the end of the introduction.

-Graphic Abstract (GA) would make the readers to follow the study easily. GA should include your your aim, methodology and results.

-in the discussion you should mention if your hypotheses were accepted or rejected.

-Expand your information regarding future studies, such as doing similar studies in different parts of the world. Comparing different type of population, gender, example races, ages, and more.

-Do not make short paragraphs with one or two sentences such as the last sentence of the introduction, first of the materials and methods, and the first like in the Discussion. You should combine that sentence with the following or previous paragraph.

Comments on the Quality of English Language

Minor revision required.

Author Response

We would like to thank the Reviewer for dedicating his/her time to our manuscript. Edits in the manuscript are highlighted in red, and below our answers:

1. Please expand the introduction, it is very short, only 3 paragraphs. You can expand the area of sustainable food such as mentioning different types of food vegetables easily growth in different parts of the world due to the different climates.

We revised the introduction. However, we preferred to add only information relevant for the study aim. 

2. Mention your hypotheses at the end of the introduction.

We mentioned our hypothesis at the begin of the introduction.

3. Graphic Abstract (GA) would make the readers to follow the study easily. GA should include your aim, methodology and results.

We added a graphical abstract.

4. In the discussion you should mention if your hypotheses were accepted or rejected.

We added this in the discussion.

5. Expand your information regarding future studies, such as doing similar studies in different parts of the world. Comparing different type of population, gender, example races, ages, and more.

We addressed this issue at the end of the discussion. 

6. Do not make short paragraphs with one or two sentences such as the last sentence of the introduction, first of the materials and methods, and the first like in the Discussion. You should combine that sentence with the following or previous paragraph.

We improved the manuscript according to this comment.